# Management of Congenital Heart Disease: State of the Art—Part II—Cyanotic Heart Defects

**DOI:** 10.3390/children6040054

**Published:** 2019-04-04

**Authors:** P. Syamasundar Rao

**Affiliations:** University of Texas-Houston McGovern Medical School, Children’s Memorial Hermann Hospital, Houston, TX 77030, USA; P.Syamasundar.Rao@uth.tmc.edu; Tel.: +713-500-5738; Fax: +713-500-5751

**Keywords:** cyanotic congenital heart defects, tetralogy of Fallot, transposition of the great arteries, tricuspid atresia, total anomalous pulmonary venous connection, truncus arteriosus, hypoplastic left heart syndrome, palliative surgery, corrective surgery, trans-catheter therapy

## Abstract

In this review management of the most common cyanotic congenital heart defects (CHDs) was discussed; the management of acyanotic CHD was reviewed in Part I of this series. While the need for intervention in acyanotic CHD is by and large determined by the severity of the lesion, most cyanotic CHDs require intervention, mostly by surgery. Different types of tetralogy of Fallot require different types of total surgical corrective procedures, and some may require initial palliation, mainly by modified Blalock–Taussig shunts. Babies with transposition of the great arteries with an intact ventricular septum as well as those with ventricular septal defects (VSD) need an arterial switch (Jatene) procedure while those with both VSD and pulmonary stenosis should be addressed by Rastelli procedure. These procedures may need to be preceded by prostaglandin infusion and/or balloon atrial septostomy in some babies. Infants with tricuspid atresia require initial palliation either with a modified Blalock–Taussig shunt or banding of the pulmonary artery and subsequent staged Fontan (bidirectional Glenn and fenestrated Fontan with extra-cardiac conduit). Neonates with total anomalous pulmonary venous connection are managed by anastomosis of the common pulmonary vein with the left atrium either electively in non-obstructed types or as an emergency procedure in the obstructed types. Babies with truncus arteriosus are treated by surgical closure of VSD along with right ventricle to pulmonary artery conduit. The other defects, namely, hypoplastic left heart syndrome, pulmonary atresia with intact ventricular septum, double-outlet right ventricle, double-inlet left ventricle and univentricular hearts largely require multistage surgical correction. The currently existing medical, trans-catheter and surgical techniques to manage cyanotic CHD are safe and effective and can be performed at a relatively low risk.

## 1. Introduction

In the first paper [1], management of acyanotic congenital heart defects (CHDs) was discussed. In this paper, discussion of most common cyanotic CHDs will be included. Cyanotic CHDs usually have multiple defects of the heart that result in right-to-left shunt. Obstruction to pulmonary blood flow (for example tetralogy of Fallot), complete admixture of pulmonary and systemic venous returns (for example, total anomalous pulmonary venous return and double-inlet left ventricle) and parallel rather than in-series circulation (transposition of the great arteries) are the usual causes of right-to-left shunts and cyanosis. In cyanotic CHD, the primary physiological abnormality is arterial desaturation due to right to left shunt across the atrial/ventricular septal defects or patent ductus arteriosus, mixing of pulmonary and systemic venous returns, or secondary to parallel circulation in transposition of the great arteries. The magnitude of arterial desaturation is dependent upon the severity of the right ventricular outflow tract obstruction or degree of intracardiac mixing. Cyanosis is a clinical sign of arterial desaturation. Chronic cyanosis results in clubbing. The chronic arterial hypoxemia increases red blood cell production by stimulating erythropoietin from the kidneys. If this is excessive, polycythemia will result. If iron sources are inadequate relative anemia results. Monitoring hemoglobin levels during follow-up of cyanotic CHD is important. The right to left shunts also predispose to cerebrovasucular accidents, strokes and brain abscesse. Long-standing hypoxemia and polycythemia result in coagulation abnormalities and hyperuricemia. These complications may be prevented by maintaining an optimal level of hemoglobin, avoiding dehydration, and by performing timely palliative or corrective surgery. With the advent of early correction of cyanotic CHD, these complications are less often seen at the present time.

The most important of the cyanotic CHDs are what are called the “5 Ts” (Table 1) and these and other important defects, as listed in Table 1, are reviewed in this paper.

Since most, if not all cyanotic CHDs need intervention, the format of this paper will be different from that used for acyanotic CHDs. For each of the cyanotic CHD, a brief description of the defect complex will be included followed by timing and methods of management concurrently; palliative and corrective therapies are discussed as appropriate for a given lesion.

Many of the cyanotic CHD patients initially present in the neonatal period. Maintaining neutral thermal environment, normal acid-base balance, normal glucose and calcium levels with suitable monitoring and correction, when needed, is essential throughout the process of recognition, transportation to a tertiary care center (if necessary), diagnostic studies including echocardiography and cardiac catheterization, as required, and therapy either with catheter interventions and/or palliative/corrective surgical procedures as well as following these procedures [2,3]. In babies with cyanotic CHD, the fraction of inspired oxygen (FIO_2_) should be 0.3 (30%) to 0.4 (40%) and 100% oxygen (O_2_) is not necessary. If ductal-dependent pulmonary or systemic blood flow is suspected, intravenous infusion of prostaglandin E_1_ (PGE_1_) (0.05 to 0.1 mcg/kg/min) should be initiated to open and/or to maintain ductal patency while waiting for confirmatory diagnosis [2,3].

The objective of management of any cyanotic CHD patients is to permit total surgical correction with negligible mortality and morbidity and to prevent or treat complications inherent to cyanotic CHD. Prevention of subacute bacterial endocarditis (SBE), and monitoring for relative anemia secondary to iron deficiency and timely treatment, when found should be undertaken. Exercise restriction is not necessary unless symptoms develop with activity.

## 2. Tetralogy of Fallot

Tetralogy of Fallot (TOF) is the most common cyanotic CHD in children beyond the age of one year [4,5]. TOF was described more than 100 years ago by Fallot [6] as a collection of four abnormalities. These include ventricular septal defect (VSD), pulmonary stenosis (PS), right ventricular (RV) hypertrophy, and dextroposition of the aorta (Ao). The VSD is almost always large and nonrestrictive and is situated in the subaortic region. Rarely, additional muscular VSDs may co-exist. The PS is variable both with regard to the location and severity of obstruction. The narrowing may be at infundibular, valvar, supra-valvar or branch pulmonary artery (PA) level or at multiple levels. The magnitude of narrowing may be very mild without any arterial desaturation, or it may be very severe causing severe cyanosis. The RV hypertrophy is typically severe in degree. The degree of over-riding of the Ao over the ventricular septum (dextroposition) varies from one patient to the next. The Ao descends on the right side in nearly 25% of cases. Atrial septal defects (ASDs) may be present in 15% of the cases and when TOF is seen with ASD, it is called pentology of Fallot.

Hypercyanotic spells may be seen in TOF patients and may be described as anoxic spells, hypoxic spells, paroxysmal hyperpnea, blue spells, tetralogy spells, paroxysmal hypoxemic spells and hypercyanotic episodes. The spells most usually occur between one and 12 months of age, with a higher frequency in two- to three-month-old babies. The spells are most common in the morning after awakening from sleep. Defecation, crying and feeding commonly precede these episodes. The spells are exhibited by elevated rate and depth of respiration and increased cyanosis, with progression to limpness and syncope. However, they usually recover. Rarely, the spells end up in convulsions, cerebrovascular accidents, or even death. The cause of the spell syndrome is unknown, but several theories have been postulated, and most possible mechanisms are right ventricular outflow tract spasm precipitated by an acute increase in catecholamines and any stimulus producing hyperpnea.

Several variants of TOF are known to exist with significant differences in anatomy, clinical presentation and the type of management necessary [4,5,7]. On the basis of these considerations, the TOF may be divided into: (1) conventional TOF, (2) TOF with pulmonary atresia, (3) TOF with multiple aorto-pulmonary collateral arteries (MAPCAs) and (4) TOF with syndrome of absent pulmonary valves [4,5,7]. Each of these entities will be addressed separately.

### 2.1. Conventional TOF

#### 2.1.1. Corrective Surgery

The majority of patients with conventional TOF are either minimally cyanotic or acyanotic in the newborn period and do not require immediate surgical intervention. Clinical follow-up and elective surgical correction between six and 12 months of age is currently the preferred approach. Total surgical correction consists of closure of VSD directing the left ventricular (LV) ejection into the Ao, and relieving the infundibular, valvar and pulmonary artery obstruction with or without trans-annular patch, all under cardiopulmonary bypass. Issues related valve-sparing surgery will be addressed by Dr. Sinha and her colleagues in a paper in this symposium.

Total correction may not possible because of pulmonary annular hypoplasia, “smallish” left ventricle, anomalous course of a major coronary artery in the right ventricular infundibulum, and/or PA hypoplasia. In such situations, palliative procedures may be performed to increase pulmonary blood flow and to allow the patients to grow to a suitable age and size so that complete correction can be safely performed later.

#### 2.1.2. Palliative Procedures

When the degree of RV outflow tract obstruction is very severe causing hypoxemia in the neonatal period, augmentation of pulmonary blood flow by prompt intravenous infusion of PGE_1_ may be undertaken. The starting dose is 0.05 to 0.1 mcg/kg/min and the dose is adjusted according to the clinical response; the PGE_1_ may be reduced stepwise downward to 0.02 to 0.025 mcg/kg/min. Most infants respond well, but a more permanent source of pulmonary blood flow should be provided since the PGE_1_ is not effective beyond the neonatal period.

Following the initial description of subclavian artery-to-ipsilateral PA anastomosis (classic Blalock–Taussig (BT) shunt) in the mid-1940s [8], a number of palliative procedures to augment pulmonary blood flow have been devised over time and these include, descending aorta-to-left PA anastomosis (Potts shunt), ascending aorta-to-right PA anastomosis (Waterston-Cooley shunt), end-to-end superior vena cava-to-right PA anastomosis (classic Glenn shunt), enlargement of the VSD, formalin infiltration of the wall of ductus arteriosus, central aortopulmonary fenestration or expanded polytetrafluoroethylene (Gore-Tex; W. L. Gore & Associates, Inc, Newark, DE, USA) graft shunt, Gore-Tex interposition graft between the subclavian artery and the ipsilateral PA (modified BT shunt), balloon pulmonary valvuloplasty, bidirectional Glenn procedure and stent implantation into the ductus [9,10]. At the present time, most surgeons favor the modified BT shunt [11] using an interposition Gore-Tex graft between right or left subclavian artery to the ipsilateral PA (Figure 1).

If the predominant obstruction is at the pulmonary valve level, though uncommon, balloon pulmonary valvuloplasty [12,13] to increase the pulmonary blood flow may be performed; balloon pulmonary valvuloplasty apart from increasing pulmonary blood flow, promotes growth and development of the pulmonary artery (Figure 2) and left ventricle so that a total surgical corrective procedure could be performed safely at a later time [12,13,14,15,16].

In premature babies two different palliative procedures were used as a bridge to total correction later. Balloon pulmonary valvuloplasty has been used in premature babies with TOF and significant arterial desaturaion as a bridge to total correction [17]. Similarly, deployment of a stent in RV outflow tract (Figure 3) may be used to promote growth of the pulmonary arteries and facilitate complete surgical correction at a later time [18].

#### 2.1.3. Management of Cyanotic Spells

Hypercyanotic or “Tet” spells may happen in infants and children prior to repair tetralogy of Fallot and in other cyanotic CHD with large inter-ventricular communication and pulmonary outflow tract obstruction. The features of and the etiology of spell syndrome was reviewed above. The management of spells may be summarized as follows [20,21,22]: the baby is placed in a knee-chest position. The cause for its effectiveness is related to increasing the systemic vascular resistance and thus decreasing the right-to-left shunt across the VSD with consequent improvement of the pulmonary flow. Humidified oxygen via a facemask is generally administered. If the child is excessively disturbed by the facemask, O_2_ may be discontinued since the major problem in the spell syndrome is pulmonary oligemia rather than alveolar hypoxia. Morphine sulfate, 0.1 mg/kg subcutaneously is effective in aborting the spell in most cases. While the mechanism of action is not clearly understood, morphine’s depressive effect on the central nervous system respiratory drive (thus reducing hyperpnea) and sedation of the infant are likely to be important. After the physical examination is finished and the limited but important laboratory studies are secured, the infant is left undisturbed and allowed to rest which may improve the infant’s condition. Correction of metabolic acidosis with sodium bicarbonate, anemia by blood transfusion, and dehydration by intravenous fluids is very important at this stage. If the spell continues, administration of vasopressors to increase the systemic vascular resistance and thus increase of the pulmonary blood flow, may be attempted. Phenylephrine may be given to increase systemic vascular resistance; the rate of infusion should be adjusted to increase the systolic blood pressure by 15% to 20% of the control value. Alternatively, propranolol, 0.1 mg/kg body weight, diluted in 50 mL of 5% dextrose in water, may be slowly administered intravenously while monitoring the heart rate. Another option is Esmolol; a loading dose of Esmolol is 500 mcg/kg is followed by maintenance doses of 50–100 mcg/kg/min. If these beta-blockers are found to be effective, the infant may be switched to oral propranolol 1 to 4 mg/kg/day in three and four divided doses. The mechanism of action of beta-blockers is not clearly understood, but several hypotheses have been put forward and include a negative inotropic effect on the RV infundibular myocardium, prevention of decrease in systemic vascular resistance and/or prevention of ventilatory response (hyperpnea) to hypoxia. If the infant does not improve with any of the aforementioned measures, general anesthesia, an emergency systemic-to-pulmonary artery shunt with modified BT anastomosis or total surgical correction should be attempted. The important principle is that the infant requires more pulmonary blood flow [20,21,22].

### 2.2. TOF with Pulmonary Atresia

In this TOF variant, the internal cardiac structure is similar to that of TOF described above, but the right ventricular outflow tract is totally obstructed. The location of atresia may be at the infundibular, valvar or main pulmonary artery level or a combination thereof. Occasionally, isolated valvar atresia may be present. The branch PAs are typically small and hypoplastic and may have discrete or diffuse narrowings. Since there is no forward flow from the RV into the PA, these patients are ductal dependent [4,5]. In the most of these infants, the ductus arteriosus is long and tortuous.

#### 2.2.1. Palliative Procedures

Since these babies are ductal dependant, all infants should receive intravenous infusion of PGE_1_ as described in the preceding section. After the diagnosis is confirmed by echo-Doppler and/or cardiac catheterization studies, as needed, a more permanent way to provide pulmonary blood flow should be instituted since the effectiveness of keeping the ductus open with PGE_1_ decreases with increasing age. As alluded to in the preceding section, a modified BT shunt [11] is a favored approach to achieve this, unless total surgical correction is feasible at presentation. Another option is to keep the ductus patent by implanting a stent into ductus arteriosus [23,24,25]. While ductal stenting is an attractive non-surgical option, generally long and tortuous ducti seen in this lesion make ductal stenting a more difficult procedure and this procedure is not currently the first-line therapeutic choice. Occasionally, the atresia is only at the valvar level without significant infundibular and PA narrowing. In such situations, perforation of the pulmonary valve followed by balloon dilatation [26,27], as is a common practice for pulmonary atresia with intact ventricular septum patients [24,28,29,30], may be undertaken.

#### 2.2.2. Corrective Surgery

Complete surgical correction with closure of VSD and pulmonary valvotomy with or without trans-annular patch or right ventricle to pulmonary artery valved conduit (usually a homograft) under cardio-pulmonary bypass is undertaken when an infant is between six and 12 months of age.

### 2.3. TOF with MAPCAs

In this TOF variant, the internal cardiac structure of the heart is similar to that of TOF with pulmonary atresia described above, without forward flow into the PA. The main PA is absent or very small. The branch PAs are small and may have multiple stenoses. Discontinuity of the right and left PA branches and of individual lobar branches may be seen. Multiple aorto-pulmonary collateral vessels supply the pulmonary blood flow. These collateral vessels, connecting the systemic and pulmonary arterial circulations, originate directly from the aorta or its branches both above and below the level of diaphragm. These collateral vessels are usually tortuous, variable in caliber and may have stenotic segments [31]. Rarely, a small patent ductus arteriosus (PDA) may be seen, but in the purest form of MAPCAs, the ductus is absent. Most of the infants with MAPCAs have reduced pulmonary blood flow with hypoxemia, although some babies, though rarely, have pulmonary over circulation.

The objective of management of TOF with MAPCAs is to restore biventricular circulation with the minimum number of surgical and catheter interventional procedures. The approach depends upon the clinical presentation (hypoxemia with O_2_ saturation less than 70%, stable O_2_ saturation between 70% and 90% and high O_2_ saturations (>90%) with congestive heart failure (CHF); these largely depend upon on quantity of effective pulmonary blood flow), pulmonary artery anatomy and surgical philosophy (multistage correction vs. total correction at presentation).

In patients with low O_2_ saturations and inadequate pulmonary blood flow, PGE_1_ infusion to open the ductus and/or an aortopulmonary shunt to augment the pulmonary blood flow may be required. In patients with adequate pulmonary blood flow with O_2_ saturations between 70% and 90% no immediate intervention is necessary. In patients with CHF, anti-congestive measures as detailed in Part 1 [1] should be instituted. Sometimes trans-catheter occlusion of a collateral vessel causing excessive pulmonary blood flow may needed (Figure 4) after ensuring that dual supply to the lung segments is present.

Once the baby is stabilized, cardiac catheterization and selective cine-angiography is performed to delineate the PA anatomy (atresia, degree of PA hypoplasia, confluence or lack of it), sources of pulmonary blood flow and degree of connections between central PAs and MAPCAs, and presence of dual supply to pulmonary arterial segments. If the native PAs are of adequate size (>50% of normal) and if the estimated RV systolic pressure following operation is less the 70% of LV systolic pressure, total correction is indicated. The latter assessments may be made by the use of McGoon ratio or Nakata index [33].

If the infant’s anatomy is not suitable for total surgical correction, staged reconstruction is contemplated. RV outflow tract is reconstructed with either a trans-annular patch or RV-to-PA conduit (usually a homograft) in an attempt to encourage growth of the native PAs. Such procedures also provide anterograde catheter access so that a more precise delineation of the pulmonary arterial tree can be undertaken and balloon angioplasty or stenting of PAs, as necessary, can be performed. Such interventions decrease RV pressures, augment flow to distal PAs and reduce ventilation/perfusion mismatch. Once adequate pulmonary blood flow is established the remaining aortopulmonary collaterals could be occluded by trans-catheter methodology, and the VSD is closed surgically.

In patients with limited number of bronco-pulmonary segments supplied by native PAs, it is essential to recruit additional segments supplied by collateral vessels, an approach described as unifocalization. In this type of surgery, the collateral vessels are disconnected from the aorta and anastomosed to the central or branch PAs as deemed appropriate. The unifocalization procedure is usually performed via thoracotomy, one side at a time. The blood supply to unifocalized PA is provided by a modified BT shunt. Incorporation of at least 14 PA segments is considered crucial to achieve a successful outcome [34].

### 2.4. TOF with Absent Pulmonary Valve Syndrome

In this TOF variant, the pulmonary stenosis is produced by hypoplasia of the pulmonary valve ring. The pulmonary valve cusps are absent or rudimentary resulting in pulmonary insufficiency. Most importantly however, there is aneurismal dilatation of the main and branch PAs causing varying degrees of tracheobronchial tree compression. Otherwise, the internal anatomy of the heart is similar to that seen with other types TOF described above, but infundibular obstruction is rare. The syndrome of absent pulmonary valve may also be seen, though infrequently, in association with other defects such as ASD, VSD, atrioventricular septal defect and double outlet RV [4,5].

Two types of clinical presentations are recognized, neonates with severe respiratory distress and patients presenting beyond six months of life with little or no respiratory symptoms. In the first group, chest physiotherapy and ventilatory support to stabilize the baby, prone position to “move the pulmonary arteries away from the tracheobronchial tree,” and reduction of the PA pressures by pulmonary vasodilators such as nitric oxide (NO) may be helpful. Babies with severe respiratory distress may require emergency sternotomy to relieve the obstruction or place the patients on extracorporeal membrane oxygenation (ECMO) [4,5,35,36,37]. Anti-congestive measures are provided in babies who have obvious signs of CHF. Once the patient is stabilized, total surgical correction to include closure of VSD, relief of pulmonary stenosis by a trans-annular pericardial patch as necessary and partial resection and plastic repair of aneurismally dilated main and branch PAs (Figure 5) should be performed under cardiopulmonary bypass. Some controversy exists with regard to whether prosthetic replacement of the pulmonary valve is required at the time of primary repair [37,38]; the author’s opinion is that valve replacement may not be necessary in all cases at the time of primary surgery [37].

The second group, the patients that present at or after six months of age, require elective correction including partial resection and plastic repair of aneurismally dilated PAs.

## 3. Transposition of the Great Arteries

Transposition of the great arteries (TGA) is the most common cyanotic CHD in the newborn. In TGA, the morphologic RV gives rise to the Ao and the morphologic LV gives rise to the PA. The atria are normal in position, i.e., atrial situs solitus, atrio-ventricular concordance (right atrium connected to the RV and the left atrium to the LV) is present, the RV is on the right and LV is on the left (d loop of the ventricles), and ventriculo-arterial discordance (aorta arising from the right ventricle and the pulmonary artery from the left ventricle) exists [21,39,40]. The aortic valve is located to the right of pulmonary valve and is characterized as d-TGA. With this anatomic arrangement, the systemic venous return from the vena cava and right atrium goes into the RV and from there into the Ao while the pulmonary venous blood goes into the left atrium and LV and from there into the PA. As a result, the circulation is parallel instead of the normal in-series circulation. Consequently, the pulmonary venous blood is not delivered to the body and the systemic venous blood is not diverted to the lungs for oxygenation. With such a circulation the neonate will not survive unless there is inter-circulatory mixing [21,39,40].

Significant differences in anatomy, clinical presentation and the type of management required mandate classification of the TGA patients into three groups [39,40]: (1) TGA with intact ventricular septum, (2) TGA with VSD, and (3) TGA with VSD and PS. Each of these types will be discussed separately.

### 3.1. TGA with Intact Ventricular Septum

In TGA with intact ventricular septum, there are no other intracardiac defects other than persistent fetal structures such a patent foramen ovale (PFO) and PDA. If there is adequate mixing across PFO and/or PDA, the infant maintains adequate O_2_ saturation; if not, the infant will present with hypoxemia. Intravenous infusion of PGE_1_ (0.05 to 0.1 mcg/kg/min), as described in the preceding sections, may help open the ductus and improve O_2_ saturation. If there is no improvement, balloon atrial septostomy (Figure 6) [41,42] is performed. In most cases, balloon atrial septostomy is successful with occasional need for blade septostomy [42,43].

Two types of surgical strategies, namely venous switch [44,45] and arterial switch [46] have been described to correct TGA. Venous switch procedures such as Senning [44] and Mustard [45] were used extensively in 1960s, 1970s and 1980s, but because of atrial arrhythmias during follow-up and leaving the right ventricle as systemic ventricle, gradually, arterial switch procedure was adopted as procedure of choice in late 1980s and there afterwards. The Jatene (arterial switch) along with the LeCompte maneuver [47] is usually performed around the age of one week. Consequently, balloon atrial septostomy may not be needed in all infants. If naturally occurring PFO and/or PGE_1_ infusion to open the ductus arteriosus do not maintain adequate O_2_ saturations (60% to 70% without metabolic acidosis), balloon atrial septostomy is required preparatory to the Jatene procedure. More recently, some groups of surgeons have been performing the Jatene procedure within the first few days of life, avoiding the need for balloon atrial septostomy altogether.

In babies presenting beyond the neonatal period, de-conditioning of the left (pulmonary) ventricle may occur. In such patients, staged correction [48,49] with initial PA banding (with or without an aortopulmonary shunt) to train the left ventricle to pump into the systemic circulation, followed by arterial switch procedure may be a better option than venous switch procedures.

### 3.2. TGA with VSD

CHF is the usual mode of presentation for patients with TGA and VSD in contradistinction to cyanosis and hypoxemia of TGA with an intact atrial septum. Anti-congestive measures should be instituted promptly, as detailed in Part 1. Relief of pulmonary venous congestion and improvement of O_2_ saturation may be accomplished by balloon atrial septostomy. Once the infant is stabilized, the Jatene procedure along with closure of VSD may be performed [48]. Such surgery may be undertaken around two to three months of age depending on control of symptomatology and adequacy of weight gain. Banding of the PA initially with subsequent complete repair is not currently recommended.

### 3.3. TGA with VSD and PS

Presentation of patients with TGA, VSD and PS is variable. If hypoxemia is secondary to poor mixing, balloon atrial septostomy should be performed. If the hypoxemia is due to decreased pulmonary flow, PGE_1_ should be infused initially followed by modified BT shunt. In patients with dominant pulmonary valvar obstruction, balloon pulmonary valvuloplasty [12,13,14,15,16] may help relieve hypoxemia.

Eventually, most of the patients in this group require a Rastelli type of repair [50] consisting of closure of VSD while redirecting the LV blood flow into the Ao and insertion of valved conduit (aortic homograft or Contegra) connecting the RV to the PA. While this type of surgery can be performed in the neonate, better results are likely when this procedure is performed between one and two years of age. Other types of surgery, namely, Kawashima (resection of infundibular septum) [51], REV (réparation étage ventriculaire) without using conduit [47] and posterior translocation of the aorta [52] have been attempted with variable degrees of success.

## 4. Tricuspid Atresia

In tricuspid atresia [53,54,55,56,57], there is absence or agenesis of the morphologic tricuspid valve; in the most common muscular type, there is a dimple or a localized fibrous thickening in the floor of the right atrium. The right atrium is enlarged. The inter-atrial communication is usually a stretched PFO. The RV is small and hypoplastic. The LV is a morphologic LV and is enlarged and hypertrophied. The VSD may be absent, small or large. Tricuspid atresia is mostly classified on the basis of associated defects [53,54,55,56,57,58]: Type I. Normally related great arteries, Type II. d-transposition of the great arteries, Type III. Malpositions of the great arteries other than d-transposition, and Type IV. Truncus arteriosus.

The management of tricuspid atresia will be discussed under the heading of palliative management and “corrective” surgery.

### 4.1. Palliative Management

Nearly 80% of tricuspid atresia patients are symptomatic in the neonatal period and require effective palliation so that they may reach the age and size at which the Fontan procedure [59,60] can be performed. The type of palliation performed is based on the hemodynamic abnormality created by tricuspid atresia and the associated cardiac lesions [9,10].

#### 4.1.1. Decreased Pulmonary Blood Flow

In neonates with markedly decreased pulmonary blood flow with low O_2_ saturations, the ductus should be made to open by intravenous infusion of PGE_1_ in a manner similar to that described in TOF and TGA sections. After the infant is stabilized and echocardiographic definition of cardiac abnormalities is undertaken, a modified BT shunt [11] is performed. Other options to increase pulmonary blood flow are enlargement of VSD and/or resection of right ventricular outflow tract obstruction [61], stenting of the ductus [23,24,25] and balloon pulmonary valvuloplasty [12,13], but modified BT is most commonly used for augmenting the pulmonary blood flow.

#### 4.1.2. Near Normal Pulmonary Blood Flow

Babies with near normal pulmonary blood flow maintaining O_2_ saturations in the low 80s do not need any intervention and may be followed clinically until the time of bidirectional Glenn procedure.

#### 4.1.3. Increased Pulmonary Blood Flow

In babies with increased pulmonary blood flow and CHF, anticongestive therapy is promptly instituted. In patients with transposition (Type II), PA banding should be performed after the infant is stabilized with anticongestive treatment. If there is associated aortic coarctation, it should also be addressed at the same time. In patients with normally related great arteries (Type I) banding of the PA should not immediately be performed because of possibility of spontaneous closure of the VSD [62,63,64,65]. If optimal anticongestive management with some waiting does not result in adequate improvement in symptoms, PA banding should be entertained; an absorbable band [66,67] may be a good option.

#### 4.1.4. Interatrial Obstruction

The entire systemic venous return must egress via the PFO, but it rarely becomes obstructive in the neonatal period, although this can be problematic later in infancy [9,10]. A right-to-left atrial mean pressure difference ≥5 mmHg with tall “a” waves (15 to 20 mmHg) in the right atrium is commonly considered to represent obstructed interatrial septum [68,69]. Such obstruction at PFO may be relieved by balloon atrial septostomy [41,42], and if unsuccessful, blade atrial septostomy [42,43,70] may have to be performed. Rarely, surgical atrial septostomy may be required to relieve the obstruction at the PFO.

#### 4.1.5. Interventricular Obstruction

Spontaneous closure of the VSD produces pulmonary oligemia in tricuspid atresia patients with normally related great arteries (Type I) and subaortic obstruction in tricuspid atresia patients with TGA (Type II) [62,63,64,65]. Management of pulmonary oligemia associated spontaneous closure of the VSD in Type I patients is the same as described in the section on “Decreased Pulmonary Blood Flow” above. In type II patients, partial spontaneous closure of the VSD produces subaortic obstruction [63,64,65] which in turn produces LV hypertrophy which poses increased risk at the time of the Fontan procedure [71]. This obstruction is generally bypassed at the time of either bidirectional Glenn or Fontan operation by anastomosis of the proximal stump of the divided PA to the ascending Ao (Damus–Kaye–Stansel) [63,64,65].

### 4.2. “Corrective” Surgery

Fontan [59], Kreutzer [60] and their associates described “corrective” surgery by different types of atrio-pulmonary anastomotic procedures without the use of RV. A number of modifications of these procedures were introduced over the years, as reviewed elsewhere [57,72,73,74]; eventually staged total cavo-pulmonary anastomosis described by de Leval and his colleagues [75] became a standard procedure along with extra-cardiac conduit diversion of the inferior vena caval blood into the PAs [76] and fenestration of the conduit [77,78]. The reason for putting corrective in quotes is because it is not truly corrective and may be better described as single ventricle palliation. The procedure is performed in three stages [56,57,79].

#### 4.2.1. Stage I

At the time of presentation, usually in the early infancy, palliation depending the physiological abnormality (modified BT shunt or pulmonary artery banding) (Figure 7) is performed after initial stabilization.

#### 4.2.2. Stage II

At about the age of six months, a bidirectional Glenn procedure [80] is performed. In this procedure, the superior vena cava (SVC) is disconnected from the right atrium and anastomosed to the PA so that the blood from SVC is directed into both branch PAs, thus the name bidirectional Glenn (Figure 8).

In patients with an additional persistent left SVC, a bilateral, bidirectional Glenn procedure (Figure 9) is performed especially if the bridging left innominate vein is small or absent.

Prior to the bidirectional Glenn procedure, normal pulmonary artery pressures should be documented either by echo-Doppler or cardiac catheterization studies. In cases with PA stenosis and mitral valve regurgitation, these abnormalities should also be repaired at the time of bidirectional Glenn procedure. Subaortic stenosis, if present, should also be bypassed by Damus–Kaye–Stansel. If there are any other hemodynamically significant abnormalities, they should also be addressed at this stage.

#### 4.2.3. Stage IIIA

Between the ages of one and four years, usually one year after the bidirectional Glenn, Fontan completion is performed by redirecting the inferior vena caval flow into the PA by either a lateral tunnel [81] or an extra-cardiac non-valved conduit [76]. Extra-cardiac conduit with fenestration (Figure 10) is preferred by most surgeons. Cardiac catheterization and selective cine-angiography is usually performed prior to this surgery to evaluate pulmonary artery anatomy and pressures, trans-pulmonary gradient, pulmonary vascular resistance (PVR), and LV end-diastolic pressure to ensure their normalcy prior to proceeding with Fontan completion. Some centers use magnetic resonance imaging (MRI) for evaluation instead of angiography. During this catheterization, significant collateral vessels, if present, are trans-catheter occluded by most cardiologists.

#### 4.2.4. Stage IIIB

The fenestration is closed by trans-catheter device implantation (Figure 11) six months to one year after stage IIIA. Occlusion of the fenestration is not necessary in all patients, although the author recommends closure to prevent arterial desaturation and paradoxical embolism.

Some babies with tricuspid atresia may have additional rare abnormalities such as vascular ring, truncus arteriosus, absent pulmonary valve syndrome, aortopulmonary window or anomalous pulmonary venous connection [74]. As reviewed elsewhere [74] initial successful palliation and eventual Fontan completion was feasible in such patients. Similarly, infants with left ventricular non-compaction also had successful Fontan procedures [74]. Therefore, the author advocates aggressive palliation while addressing the associated cardiac abnormality and eventual staged Fontan completion irrespective of associated abnormalities.

## 5. Total Anomalous Pulmonary Venous Connection

In total anomalous pulmonary venous connection (TAPVC), all the pulmonary veins drain into a common pulmonary vein which in turn is connected to the left innominate vein, SVC, coronary sinus, portal vein or other rare sites [82,83]. Occasionally, the individual pulmonary veins may be connected directly to the right atrium or to its tributaries (mixed type). Eventually both the pulmonary venous and systemic venous blood flows return into the right atrium and from there are redistributed to the systemic circuit via the PFO and to the pulmonary circuit via the tricuspid valve. The TAPVC is generally classified on the basis of the anatomic location to which the common pulmonary vein is connected, namely, supra-diaphragmatic or infra-diaphragmatic types. It is also classified on the basis of physiology, namely, obstructive or non-obstructive. The supra-diaphragmatic types are usually non-obstructive although on rare occasion obstruction can be found, as reviewed earlier [84]. However, the infra-diaphragmatic TAPVC is almost always obstructive. Infra-diaphragmatic type is the most common form in the neonatal population while TAPVC connected to the left innominate vein is the most common type beyond the neonatal period.

In the non-obstructive type, following control of CHF, elective or semi-elective surgical correction is recommended. Since the entire systemic flow must pass through the PFO, it may become restrictive and in such situations balloon atrial septostomy is helpful [42]. However, surgical correction is the main approach. Surgical repair consists of anastomosing the common pulmonary venous confluence with the left atrium under cardiopulmonary bypass and/or hypothermia. In infants with TAPVC draining into coronary sinus, surgical removal of the common wall between the coronary sinus and left atrium is carried out along with closure of coronary sinus opening into the right atrium. Most surgeons perform suture closure of PFO and ligate the connecting vein.

In the obstructive type, the baby should be stabilized by intubation and ventilation with high airway pressure. PGE_1_ infusion may beneficial by opening the ductus and decompressing the pulmonary circuit and may potentially open the ductus venosus, thus decreasing pulmonary venous obstruction. However, urgent correction by anastomosis of the common pulmonary vein with the left atrium should be sought.

In mixed type of TAPVC, there is no large common pulmonary venous confluence and therefore, conventional surgery described above is not feasible. Unless there is evidence for pulmonary venous obstruction, waiting until anomalous veins are large enough to be anastomosed to the left atrium is most appropriate [85].

## 6. Truncus Arteriosus

Truncus arteriosus is a cyanotic CHD in which by a single arterial vessel (truncus) originates from the heart which gives rise to the Ao, PAs and coronary arteries [86,87,88]. A large VSD is present in the conal ventricular septum and the truncus overrides the ventricular septum. The PAs originate from the truncus and the nature of branching pattern forms the basis of Collette and Edwards classification [89]: Type I: a short segment of main PA originates from the left-posterior aspect of the truncus and divides into right and left PAs, Type II: the main PA is absent and both branch PAs arise close to each other, from the posterior part of the truncus, Type III: the main PA is absent and both branch PAs arise from either sides of the truncus, and Type IV: the main PA is absent and both branch PAs arise from descending aorta. The type IV is also called “pseudo-truncus” is no longer considered a type of truncus arteriosus, but as a type of pulmonary atresia with VSD. The truncal valve is usually competent and without stenosis, although on occasionally significant stenosis or regurgitation may be seen.

The treatment of choice for truncus arteriosus is total surgical correction under cardiopulmonary bypass [90] and is currently performed in the neonatal period, around the age of 10 days following control of CHF, if present. Two stage repair with banding of the PA in infancy followed by total surgical correction in childhood is no longer practiced. Surgery consists of closure of VSD, separation of the PAs from the truncus and anastomosing them to the RV via aortic or pulmonary homograft and closing the defect in the truncus produced by disconnecting the PAs. The truncal valve does not typically require surgery unless there is significant truncal valve stenosis or regurgitation. Associated defects such as interrupted aortic arch should also be repaired at the same time.

## 7. Hypoplastic Left Heart Syndrome

Hypoplastic left heart syndrome (HLHS) is a cyanotic CHD consisting of diminutive left ventricle with under developed mitral and aortic valves [91,92,93,94,95,96]. The left atrium is generally small and hypoplastic. The mitral valve is severely stenotic, hypoplastic or atretic. The left ventricular cavity is usually a slit-like structure and the LV muscle is thick. The ventricular septum is most usually intact without a VSD. The aortic valve is atretic or severely narrowed. The ascending Ao is also very hypoplastic with diameters varying from 2 to 3 mm, serving as a conduit to supply coronary blood flow in a retrograde fashion. The interatrial septum is usually thick with a PFO allowing blood egress from the left to right atrium; it may be restrictive or infrequently the atrial septum is intact. A PDA is present allowing flow from the main PA to the descending Ao and is essential for infant’s survival, but tends to close with time. Coarctation of the Ao may be seen in some of these babies. Hypoplasia of the left ventricle may also be seen in patients with double-outlet right ventricle with mitral atresia, unbalanced atrio-ventricular septal defects and other forms of complex CHD and these variants account for as many as 25% of all HLHS infants in some studies [92,97].

At the present time, there are two options to treat HLHS and these are the multistage Norwood procedure [97,98,99] and heart transplantation [100]. Due to scarcity of cardiac donors for transplantation, multistage palliation has become the mainstay in management of HLHS. The management will be discussed under the following headings.

### 7.1. Prenatal Identification

Due to routine obstetric ultrasound screening studies, a significant number of HLHS babies are identified prenatally. Such prenatal identification allows time for counseling parents and planning of delivery at a tertiary care hospital equipped to treat complex CHD; the latter helps avoid transport-related morbidity and mortality. Immediately after the baby is born, PGE_1_ infusion should be initiated to maintain ductal patency and an echocardiogram performed to confirm the diagnosis of HLHS and to appraise the adequacy of shunting across the PFO.

### 7.2. Postnatal Presentation

Neonates who are not diagnosed prenatally are usually born at term and appear normal at first. As the ductus arteriosus starts to constrict, generally over the first 24 to 48 h after birth, symptoms such as tachypnea, respiratory distress, cyanosis, pallor, lethargy, metabolic acidosis and oliguria appear. Without immediate reopening the ductus arteriosus by PGE_1_, death rapidly ensues. The same symptoms may also be found if an abrupt drop in pulmonary vascular resistance (PVR) occurs. Infants with restrictive PFO, though rare, present with respiratory symptoms and severe cyanosis secondary to pulmonary venous obstruction.

### 7.3. Management at Presentation

Management prior to surgical intervention involves three major principles, namely: (1) maintain sufficient systemic blood flow by keeping the ductus open, (2) limit excessive pulmonary flow, particularly by balancing with systemic circulation and (3) ensure adequate egress of the left atrial blood.

#### 7.3.1. Maintain Systemic Flow

The systemic circulation in infants with HLHS completely depends on the patency of ductus arteriosus. Consequently, intravenous infusion of PGE_1_ should be started as soon as the diagnosis of HLHS is made/suspected so that ductal patency is established, and adequate systemic perfusion is provided. As mentioned in the preceding sections, an initial dose is 0.05 to 0.1 mcg/kg/min is administered. After ductal patency is established, the rate of PGE_1_ infusion may be gradually (stepwise) reduced down to 0.02 mcg/kg/min. It is important to recognize that ductal patency should be maintained with the lowest effective dose of PGE_1_ so that the dose-dependent side effects of PGE_1_ such as hypotension, prompting volume resuscitation and respiratory depression and apnea, requiring mechanical ventilatory support, are minimized.

#### 7.3.2. Limit Excessive Pulmonary Flow

The PVR is somewhat higher than the systemic vascular resistance (SVR) at birth, but begins to fall after birth. In babies with HLHS, as the PVR decreases, progressive increase in pulmonary blood flow takes place with concurrent decrease in systemic blood flow. This will result in systemic hypo-perfusion and consequent metabolic acidosis and shock. Subsequent to infusion of PGE_1_ and establishing ductal patency, attempts should be made to balance systemic and pulmonary resistances by decreasing the SVR or augmenting the PVR or a combination thereof. If the infant is stable with O_2_ saturations in the mid 80s, it is prudent to just observe the baby and not administer supplemental O_2_. If the systemic O_2_ saturations trend upwards, intervention is appropriate. Maneuvers to increase PVR are more effective. Intubation and controlled ventilation with sedation and paralysis prevents hyperventilation and allow hypoventilation to keep arterial PCO_2_ between 45 and 50 mmHg. Metabolic acidosis should be corrected with sodium bicarbonate and the hematocrit maintained between 40% and 45% to provide adequate oxygen carrying capacity and to increase the blood viscosity; the latter may also assist in increasing PVR. Supplemental O_2_ should be avoided at all costs. If the above described methods fail to achieve balancing of SVR/PVR, sub-ambient O_2_ (FIO_2_ of 15% to 19%) with supplemental nitrogen or carbon dioxide may be utilized to increase PVR and permit larger systemic blood flow. Although this is an attractive concept, it should not be applied for long time periods since severe pulmonary hypertension is likely to adversely influence the postoperative recovery. However, such therapy does not seem to adversely influence the pulmonary vasculature on long-term follow-up [101].

#### 7.3.3. Ensure Adequate Egress of the Left Atrial Blood

The left atrial blood must egress into the right atrium, across the PFO, because of the obstructed mitral valve/LV. In some infants, the PFO may be small at birth or may become restrictive with time. Mild degree of narrowing may be beneficial in that it will increase the PVR which in turn may help adequate systemic flow. However, severe restriction produces pulmonary edema with consequent hypoxemia. Consequently, it is important to perform periodic echo-Doppler studies in order to monitor the size of the PFO and the degree of restriction. When the obstruction becomes severe, enlargement of the PFO by trans-catheter techniques [41,42,43,70] should be performed. While Rashkind’s balloon atrial septostomy [41] and Park’s blade septostomy [43,70] are conventional methods to lessen atrial septal obstruction, such techniques are not easy to perform because of the hypoplastic left atrium. Static balloon dilatation of the atrial septum [42,102,103] with a balloon angioplasty catheter (Figure 12) is useful. Such a procedure may not only help relieve severe obstruction, but also retain a mild degree of obstruction so that rapid fall in the PVR does not occur. Rarely, the obstruction is extremely severe and may require stent implantation [42,104].

#### 7.3.4. Other Measures

The objective of pre-operative management is to keep the babies in room air with O_2_ saturations in the 70s and 80s (by pulse oximetry). Inotropic drugs may be administered only in severely ill babies with severe CHF, acidosis, concomitant sepsis, or cardiogenic shock. Since the inotropic agents negatively affect the balance between the PVR and SVR, these drugs should be weaned off as soon as the baby becomes stable. Diuretics may be given to control pulmonary over circulation prior to Norwood operation. It is important to recognize that the status of PVR changes quickly and frequent monitoring of these infants until palliative surgery is critical.

### 7.4. “Corrective” Surgery

The objective of the management of infants with HLHS is to ultimately separate the pulmonary and systemic circulations by multistage surgical reconstruction [97,98,99]. As only one functioning ventricle is present, the right ventricle is allowed to stay as the systemic ventricle while the blood passively flows into the lungs by using the Fontan principle [59,60]. Since the PA pressure and PVR are high in the neonate, the Fontan procedure cannot be performed in the neonatal period. Therefore, eventual Fontan is achieved in three stages: (1) Stage I, Norwood procedure, (2) Stage II, bidirectional Glenn procedure, and (3) Stage III, Fontan completion.

#### 7.4.1. Stage I (Norwood Procedure)

In the Norwood procedure [97,98,99], the PA and the Ao are anastomosed with the use of additional prosthetic material as needed, atrial septectomy is performed to ensure unrestricted blood flow across the atrial septum, the ductus arteriosus is ligated, coarctation of the aorta, if any, is eliminated and an aorta-to-pulmonary artery shunt (usually a modified BT shunt) is created to supply pulmonary blood flow (Figure 13). Thus, the systemic circulation is supported by the RV and the pulmonary circulation is provided by the modified BT shunt. The Norwood procedure is usually performed around the age of one week, after stabilization of the infant as detailed above.

Some of the poor results following Norwood have been attributed to lower diastolic coronary perfusion pressures and to address this problem, Sano and his associates [105,106] replaced modified BT shunt with a RV outflow tract to PA Gore-Tex graft (Figure 13C); this procedure is now called the Sano modification of Norwood. Comparison of these two procedures among contemporary patient groups did not discover any advantage of one technique over the other [107,108]; however, most centers seem to favor the Sano modification of Norwood.

#### 7.4.2. Stages II (Bidirectional Glenn), IIIA (Fontan Completion) and IIIB (Fenestration Closure)

These stages are similar to those described in the “Tricuspid Atresia” section and will not be repeated (Figure 8, Figure 9, Figure 10 and Figure 11). However, it should be mentioned that there is considerable inter-stage mortality (5–15%) [22,109,110] which may be related to restrictive atrial communication, obstruction of the aortic arch, shunt blockage, distortion of the pulmonary arteries, atrio-ventricular valve insufficiency or a combination thereof [110]. Common childhood gastrointestinal or respiratory illnesses cause hypovolemia and/or hypoxemia have been blamed to cause inter-stage mortality [22,110]. After successful Norwood palliation, ongoing vigilance is necessary well beyond the early postoperative period in order to reduce such inter-stage mortality [22].

### 7.5. Cardiac Transplantation

Transplantation of the heart is another surgical alternative [100]. After management at presentation, as reviewed in the preceding section, the infant should be continued on PGE_1_ to maintain the ductus arteriosus patent while waiting for a donor heart. Nearly 20% of the babies listed for transplantation die while waiting for the heart to become available. Following successful heart transplantation, multiple medications for prevention of graft rejection are required. Frequent outpatient visits and periodic endomyocardial biopsy to identify rejection very early are important in the management of these children. Scarcity of cardiac donor hearts has forced the majority of institutions to perform the Norwood procedure initially followed by staged Fontan [7,94,95,96].

### 7.6. Alternative Approaches

Due to relatively high morbidity and mortality rates following the Norwood procedure, a hybrid approach [111] was suggested. In this procedure, banding of both the branch pulmonary arteries via median sternotomy is performed with concurrent implantation of a stent in the ductus arteriosus via pulmonary arteriotomy. This may be considered Stage I. During the second stage, aortic arch reconstruction, atrial septectomy and bidirectional Glenn shunt are undertaken. Stage III is Fontan completion. There appears to be a reduction in Stage I mortality, but some of it seems to be shifted to Stage II. Comparison of hybrid with conventional Norwood did not demonstrate any improvement in outcome with the hybrid procedure [112]. Additionally, there was lower systemic and cerebral oxygen delivery with hybrid than Norwood [113]. It would appear that most centers reverted back to conventional Norwood with Sano.

Simple bilateral pulmonary artery banding without ductal stent, but with continuation of PGE_1_ to keep the ducts open has been used in premature infants with a plan for Norwood later with varying degrees of success [96].

## 8. Pulmonary Atresia with Intact Ventricular Septum

Pulmonary atresia with intact ventricular septum is characterized by total obstruction of the pulmonary valve, two separate ventricles, a patent, though small tricuspid valve, and no VSD [29,114,115]. The right ventricle is usually small and hypoplastic, although on occasion, it may be close to normal in size. A PFO is usually present and facilitates right to left shunt, decompressing the right atrium. The PAs are usually well formed in contradistinction to TOF with pulmonary atresia. Coronary arteriovenous fistulae are seen in 10–50% patients. When atresia or severe stenosis of the coronary arteries is associated with these fistulae, RV dependent coronary circulation occurs; such patients are not candidates for relief of RV outflow tract obstruction.

The objective of management plans is to accomplish a four-chamber, biventricular, completely separated pulmonary and systemic circulations [7,29,114,115]. Such objective may be achieved if there is no right ventricular-dependent coronary circulation, severe right ventricular hypoplasia does not exist, and infundibular atresia is not present. Since these patients have ductal dependent pulmonary circulation, PGE_1_ infusion should be started immediately and careful assessment of the anatomy by echocardiography and if necessary, by cardiac catheterization and selective cineangiography, should be made.

If the RV is of reasonable size and tripartite or bipartite in nature (Figure 14), trans-catheter perforation with radiofrequency wire (Figure 15) followed by balloon valvuloplasty is performed to restore RV to PA continuity (Figure 16 and Figure 17).

An alternative approach used in some centers is surgical pulmonary valvotomy. Surgical pulmonary valvotomy along with RV outflow tract patch may also be performed. Some other centers use the hybrid approach; perforation of the atretic pulmonary valve is undertaken via a needle introduced through RV outflow tract exposed by sternotomy. Serial balloon dilatations of the pulmonary valve follow [115,117].

If the RV is very small (Tricuspid valve Z score ≤2.5) or right ventricular-dependent coronary circulation exists, modified Blalock–Taussig shunt along with atrial septostomy should be undertaken. Stenting of the ductus [23,24,25] is another option. These patients will then become candidates for Fontan. In some institutions, cardiac transplantation is considered for these patients [115].

If the RV size is somewhere in between, one and one-half ventricular repair should be considered. In this method of treatment, the marginal-sized RV is allowed to pump into the PA, the PFO is closed and bidirectional Glenn shunt performed. The RV is likely to tolerate pumping smaller quantities of blood because of removing SVC flow by bidirectional Glenn.

Orderly and well-planned treatment algorithms on the basis of RV size and other factors as alluded to above may help improve survival rate [24,29,114,115].

## 9. Double-Outlet Right Ventricle (DORV)

In DORV both the PA and Ao arise from the morphologic RV [118,119,120,121]. A large VSD is usually present and is the sole exit for the LV. The VSD is usually large, but may become smaller and spontaneously close after birth, resulting in severe LV obstruction [62]. DORV is classified on the basis of the location of the VSD and presence of PS [119,120]. In the most common type (nearly 50%), the VSD is in the subaortic, perimembraneous region and directs the LV flow into the Ao. If there is no PS, the clinical features resemble those of large isolated VSD. If significant PS is present, the clinical picture is similar to that of TOF. If the VSD is in the sub-pulmonary region (close to 25%), the LV flow is mostly directed into the PA, resembling the features of TGA and is generally referred to as the Taussig–Bing anomaly. In the final 25% cases, the VSD is either doubly committed to both great vessels or not committed to either Ao or PA. The size and the position of the VSD and the extent of PS determine the pathophysiology and clinical features.

In babies with subaortic VSD without PS, surgical closure of the VSD in such a way as to divert the LV blood into the Ao is performed. Due to high PVR, these babies usually do not manifest CHF until a few weeks of life. If they present in severe CHF, appropriate anticongestive measures are instituted prior to surgery. Initial pulmonary artery banding with later surgical correction is not used in the current day practice. In babies with subaortic VSD and PS, surgical correction is the same as that used for TOF repair, including relief of subpulmonary and valvar obstruction, but ensuring the VSD patch is placed in such manner to avoid LV outflow tract obstruction. If the anatomic arrangement is not suitable for surgery in the early infancy, a modified BT shunt [11] is performed initially with a plan for total correction later.

In patients with a Taussig–Bing anomaly (subpulmonary VSD), an arterial switch procedure [46] along with VSD closure is the current approach. If there is associated aortic coarctation, it should also be repaired at the same time. Again, initial PA banding is not recommended unless there are extenuating circumstances. Sometimes, balloon atrial septostomy to relieve hypoxemia may be needed while awaiting for surgical correction. If there is associated PS, the Rastelli procedure [50] is necessary.

DORV patients with uncommitted VSD are indeed problematic and usually require individualized plans to either correct or palliate these children.

## 10. Double-Inlet Left Ventricle and Univentricular Hearts

In double-inlet left ventricle (DILV), both atrioventricular valves empty into the ventricle which most often has LV morphology [7,120,121]. An outlet chamber is attached to this ventricle which has RV morphology. The atrioventricular valves may be normal, stenotic, hypoplastic or one of them may be even atretic. The great vessels are most frequently transposed with the aorta arising from the hypoplastic RV chamber and the PA from the LV. L-transposition is more frequent than d-transposition and normally related great arteries is seen in less than 30% of cases. Double-outlet RV may also be seen where both great arteries come off the rudimentary RV chamber. Two-thirds of the patients have PS. The obstruction may be valvar or subvalvar in location [7,120,121]. Pulmonary atresia may also be seen in some cases. Subaortic obstruction may manifest in patients with transposition and is secondary to narrowing of the bulbo-ventricular foramen (VSD) [120,121,122]. Such patients frequently have aortic coarctation. The phrase univentricular heart is used to characterize any cardiac lesion with one functioning ventricle and includes DILV, single ventricle, common ventricle and univentricular atrioventricular connection.

The pathophysiology produced by multiple defects in the above two entities is similar to that described for tricuspid atresia and consequently the management is similar [7,120,121]. The general objective is to accomplish Fontan circulation by staged total cavopulmonary connection. In young babies with decreased pulmonary blood flow, a modified Blalock–Taussig shunt (Figure 7A) is performed and if the pulmonary blood flow is increased, PA banding (Figure 7B) is undertaken. Relief of aortic coarctation, if present, should be provided. Subsequently, bidirectional Glenn (Figure 8 and Figure 9) and Fontan completion (Figure 10 and Figure 11) are undertaken as described in the “Tricuspid Atresia” section. Similarly, subaortic obstruction should be addressed by bypassing the obstruction by Damus–Kaye–Stansel, again as described in the “Tricuspid Atresia” section.

## 11. Other Defects

Discussion of management of other cardiac defects such as coronary artery anomalies [123] including anomalous origin of left coronary artery from the PA [124], interrupted aortic arch [125], Ebstein’s anomaly of the tricuspid valve [126,127,128,129], mitral atresia with normal aortic root [121,130], corrected transposition of the great arteries [121] and dextrocardia and heterotaxy syndromes (Asplenia and polysplenia) [131,132,133,134,135], was not included either in Part I or II of these reviews because of space limitations, but the interested reader may review the cited references at their discretion.

## Figures and Tables

**Figure 1 children-06-00054-f001:**
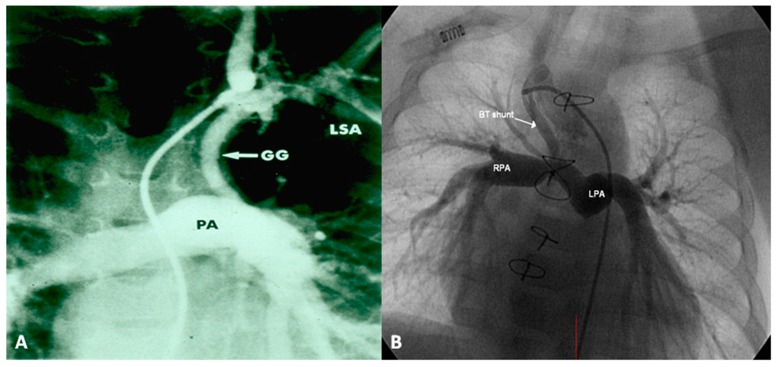
Selected frames from cine-angiograms of Gore-Tex grafts (GG) of modified Blalock–Taussig (BT) shunts in two different patients demonstrating a wide-open BT shunt with good opacification of the pulmonary artery (PA) (**A**) and of the right (RPA) and left (LPA) pulmonary artery (**B**).

**Figure 2 children-06-00054-f002:**
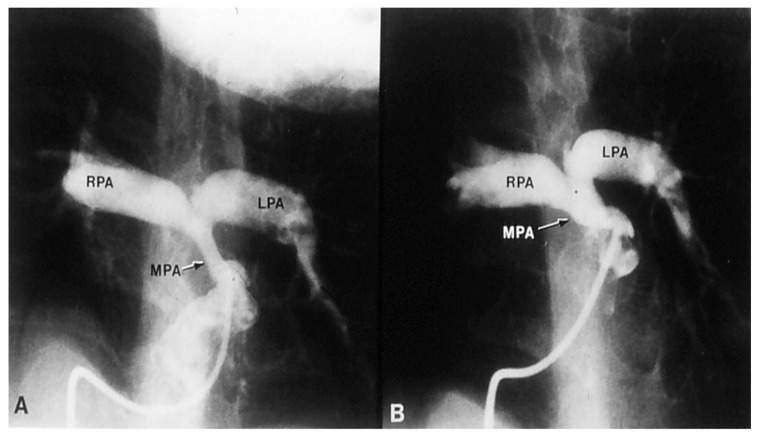
Selected frames from a pulmonary artery cineangiogram in a sitting-up view in a patient with tetralogy of Fallot prior to (**A**) and 12 months following (**B**) balloon pulmonary valvuloplasty. Note significant improvement in the size of the valve annulus and main and branch pulmonary arteries at follow-up. LPA, left pulmonary artery; MPA, main pulmonary artery; RPA, right pulmonary artery. Reproduced with permission from Rao, P.S.; et al. *Cathet Cardiovasc. Diagn.* 1992, *25*, 16–24 [13].

**Figure 3 children-06-00054-f003:**
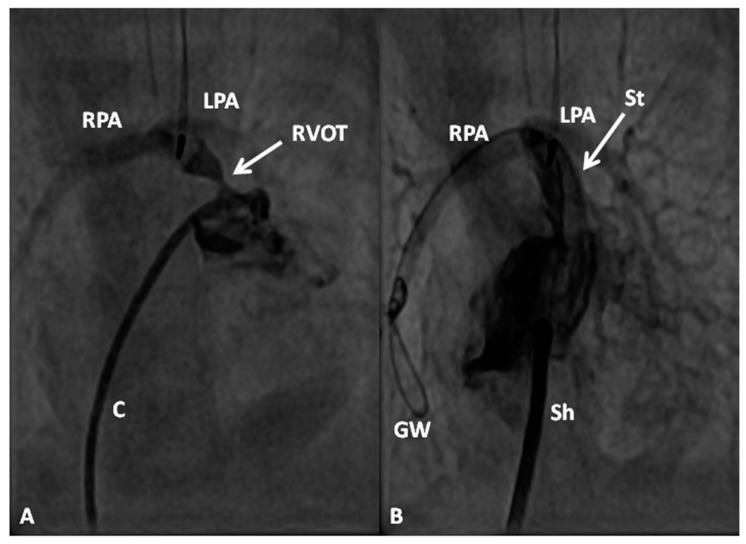
Selected cine-angiographic frames from left anterior oblique views demonstrating narrowed right ventricular outflow tract (RVOT) (arrow) prior to (**A**) and immediately following (**B**) stent (St) implantation across the RVOT. C, catheter; GW, guide wire deep in the right pulmonary artery (RPA); LPA, left pulmonary artery; Sh, sheath. Reproduced from Rao, P.S. Stents in the management of vascular obstructive lesions associated with congenital heart disease. In *Cardiac Catheterization and Imaging (From Pediatrics to Geriatrics)*; Vijayalakshmi, I.B, Ed.; Jaypee Publications: New Delhi, India, 2015; pp. 573–598 [19].

**Figure 4 children-06-00054-f004:**
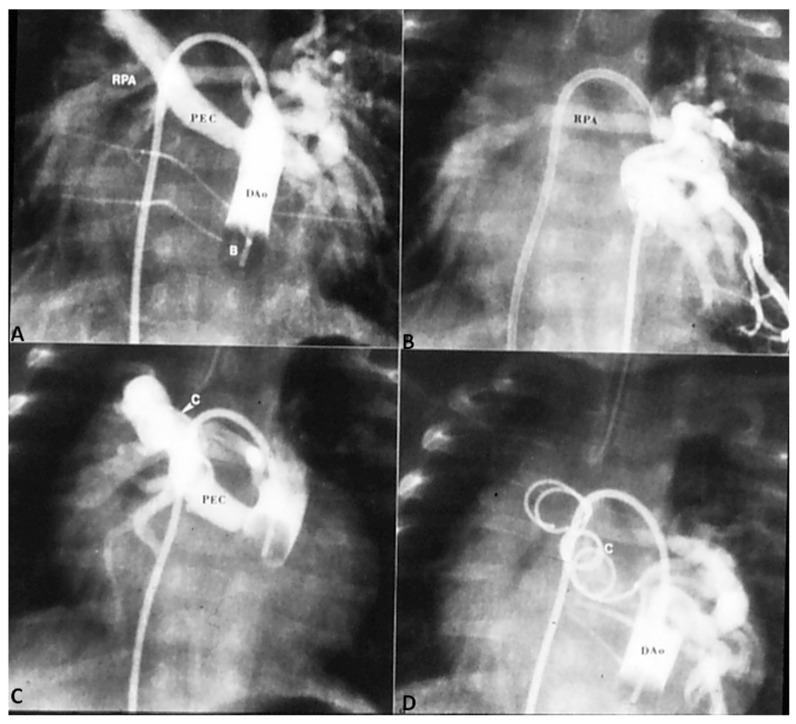
(**A**) Selected cineangiographic frame from balloon (**B**) occlusion aortography in a neonate with severe congestive heart failure demonstrating a large persistent embryonic collateral (PEC) vessel connecting the descending aorta (DAo) to the right pulmonary artery (RPA). (**B**) Alternative blood supply to RPA is demonstrated from a PEC on the left side. (**C**) Occlusion with an 8-mm diameter Gianturco coil (**C**) did not result in complete occlusion of the PEC. (**D**) A second overlapping coil was placed which resulted in complete occlusion of the PEC. The RPA was opacified via the collateral vessels from the left side. The infant improved remarkably from congestive heart failure and eventually underwent unifocalization and complete correction. Reproduced from Rao, P. S. Transcatheter embolization of unwanted blood vessels in children. In *Catheter Based Devices for Treatment of Noncoronary Cardiovascular Disease in Adults and Children*; Rao, P.S., Kern, M.J., Eds.; Lippincott, Williams & Wilkins: Philadelphia, PA, USA, 2003; pp. 457–473 [32].

**Figure 5 children-06-00054-f005:**
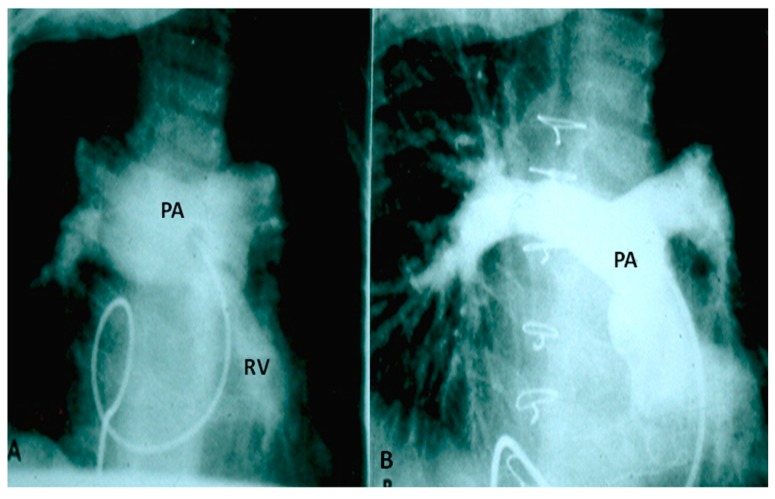
(**A**). Selected frame from a right ventricular (RV) cine-angiogram in a patient with syndrome of absent pulmonary valve obtained before surgery demonstrating marked dilatation of the main, right and left pulmonary arteries (PA). (**B**) An angiographic frame showing the size of the pulmonary arteries following the operation. The catheter tip is located at the junction of the RV with the main PA. This cine was obtained at a higher magnification factor (6″) than the pre operative film (9″). Modified from Rao, P.S.; Lawrie, G.M. *Br. Heart J.*
**1983**, *50*, 586–589 [37].

**Figure 6 children-06-00054-f006:**
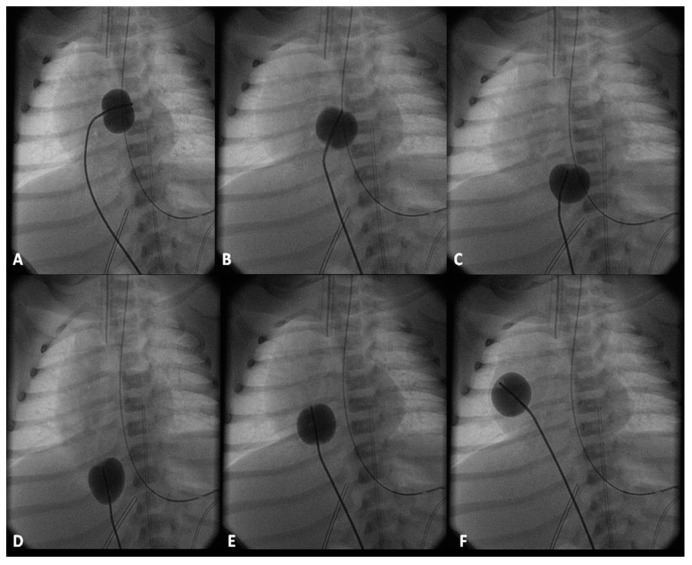
Selected cinfluroscopic frames of the Rashkind’s balloon septostomy procedure. Note the position of the inflated balloon in the left atrium (**A**) and in the right atrium and inferior vena cava in successive frames, as it is rapidly and forcefully withdrawn across the atrial septum (**B**–**D**). After it reaches the inferior vena cava (**D**), it is rapidly advanced into the right atrium (**E**) and (**F**) in order not to inadvertently occlude the inferior vena cava in case of failure to deflate the balloon (which is quite rare). Reproduced from Rao, P.S. *Neonatol. Today*
**2010**, *5(8)*, 1–8 [42].

**Figure 7 children-06-00054-f007:**
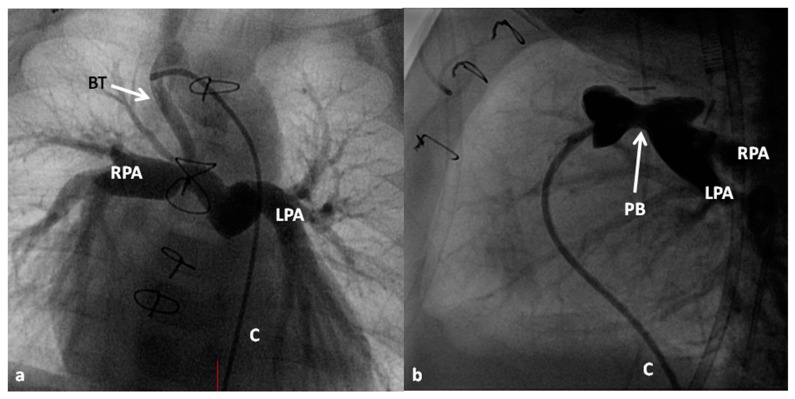
Selected cineangiographic frames demonstrate Blalock–Taussig (BT) shunt (arrow in (**a**) to treat babies with pulmonary oligemia (**a**) and pulmonary artery banding (PB) (arrow in (**b**)) in infants with pulmonary plethora (**b**) during Stage I of Fontan procedure. C, catheter; LPA, left pulmonary artery; RPA, right pulmonary artery. Reproduced from Rao, P.S. *Indian J. Pediatr.* 2015, *82*, 1147–1156 [79].

**Figure 8 children-06-00054-f008:**
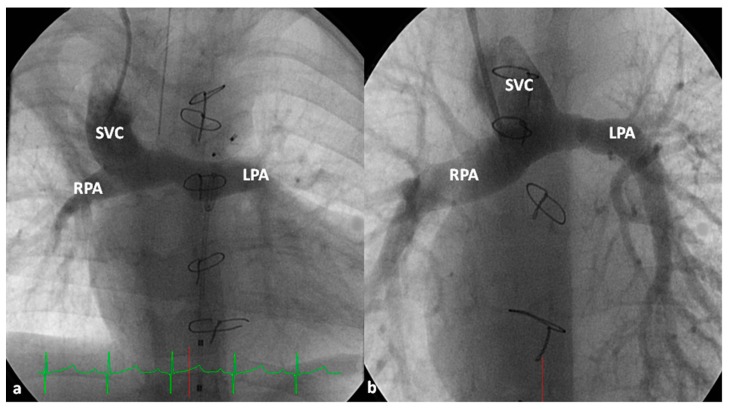
Selected cineangiographic frames representing bidirectional Glenn procedure (anastomosis of the superior vena cava (SVC) to the right pulmonary artery (RPA)) in two different children (**a** and **b**) (Stage II). Note the unobstructed flow from the SVC to the right (RPA) and left (LPA) pulmonary arteries. Reproduced from Rao, P.S. *Indian J. Pediatr.*
**2015**, *82*, 1147–1156 [79].

**Figure 9 children-06-00054-f009:**
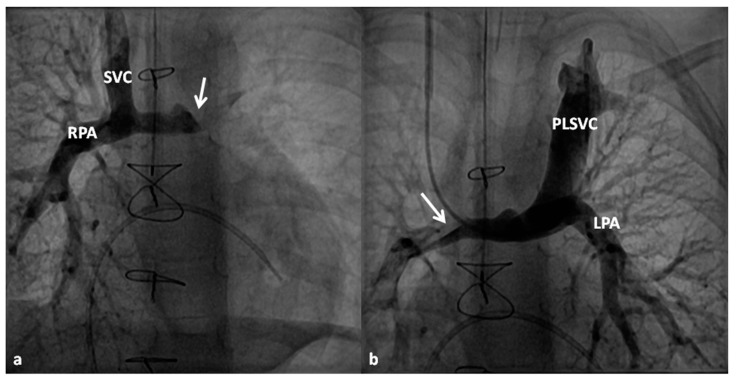
Selected cineangiographic frames showing bilateral bidirectional Glenn procedure (Stage II). In (**a**), injection into the superior vena cava (SVC) shows prompt opacification of the right pulmonary artery (RPA). The arrow in (**a**) points to the un-opacified blood from persistent left superior vena cava (PLSVC). In (**b**), injection into the PLSVC shows prompt opacification of the left pulmonary artery (LPA). The arrow in (**b**) points to the un-opacified blood from the right SVC. Note unobstructed flow from the respective SVCs into the pulmonary arteries. Reproduced from Rao, P.S. *Indian J. Pediatr.*
**2015**, *82*, 1147–1156 [79].

**Figure 10 children-06-00054-f010:**
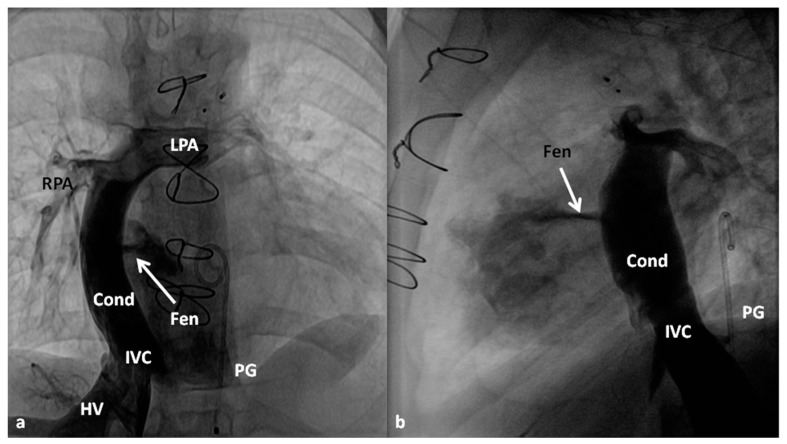
Selected cine frames in posterio-anterior (**a**) and lateral (**b**) views demonstrating Stage IIIA Fontan diverting the inferior vena caval (IVC) flow into the pulmonary arteries via a non-valved conduit (Cond). Flow across the fenestration (Fen) is shown by arrows in (**a**) and (**b**). HV, hepatic veins; LPA, left pulmonary artery; PG, pigtail catheter in the descending aorta; RPA, right pulmonary artery. Modified from Rao, P.S. *Indian J. Pediatr.*
**2015**, *82*, 1147–1156 [79].

**Figure 11 children-06-00054-f011:**
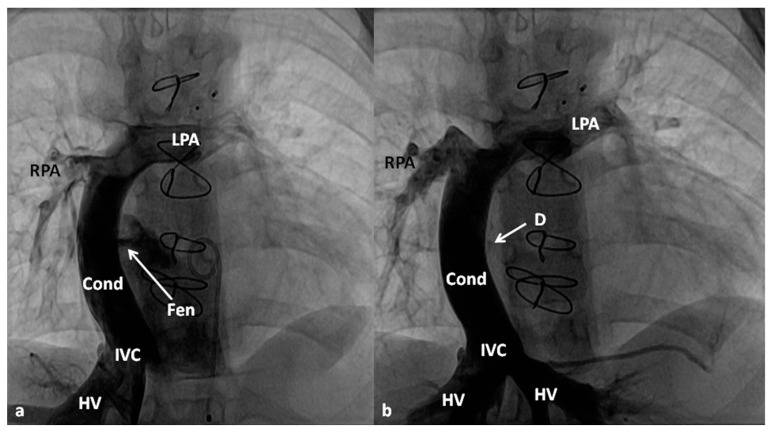
Selected cineangiographic frames in posterio-anterior view demonstrating Stage IIIA Fontan diverting the inferior vena caval (IVC) flow into the pulmonary arteries via a non-valved conduit (Cond). Note the fenestration (Fen); arrow in (**a**). The Fen is closed with an Amplatzer device (D); arrow in (**b**) (Stage IIIB). HV, hepatic veins; LPA, left pulmonary artery; RPA, right pulmonary artery. Reproduced from Rao, P.S. *Indian J. Pediatr.*
**2015**, *82*, 1147–1156 [79].

**Figure 12 children-06-00054-f012:**
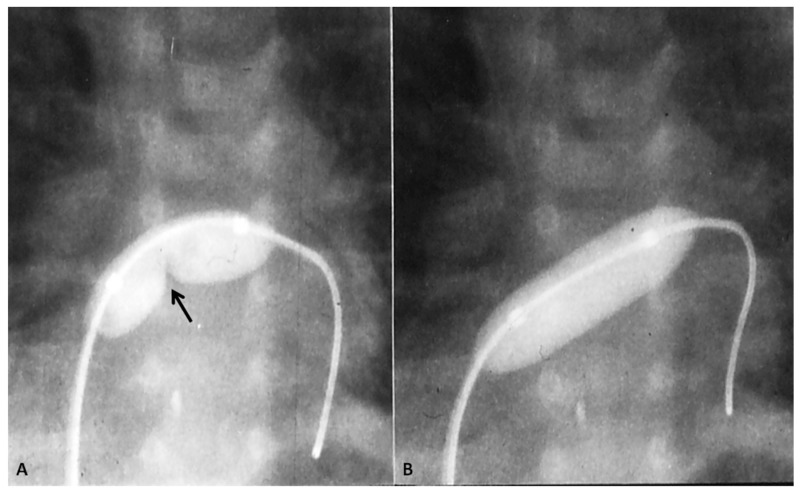
Selected cinefluoroscopic frames of balloon angioplasty procedure (to enlarge the PFO) demonstrating an inflated balloon in the posterio-anterior (**A**) and (**B**) views showing waisting of the balloon (arrow in (**A**)), which was completely abolished following further inflation of the balloon (**B**).

**Figure 13 children-06-00054-f013:**
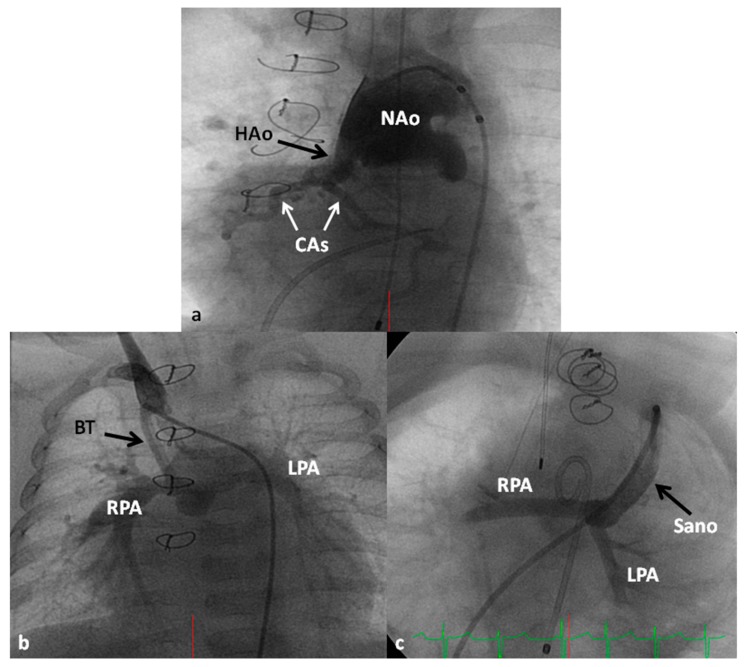
Selected cineangiographic frames illustrating the Norwood procedure with neoaorta (NAo) and hypoplastic aorta (HAo) supplying the coronary arteries (CAs) in (**a**) are shown. Blalock–Taussig (BT) shunt in (**b**) and Sana shunt in (**c**) from two different babies are also presented; this is Stage I of Fontan for babies with hypoplastic left heart syndrome. LPA, left pulmonary artery; RPA, right pulmonary artery. Reproduced from Rao, P.S. *Indian J. Pediatr.*
**2015**, *82*, 1147–1156 [79].

**Figure 14 children-06-00054-f014:**
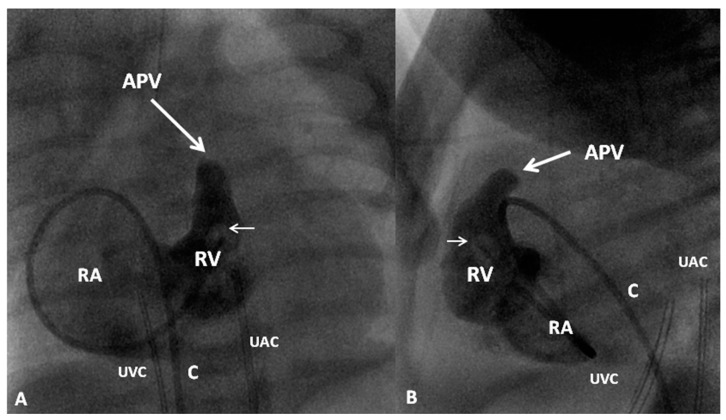
Selected cine frames from right ventricular (RV) cineangiogram in posterio-anterior (**A**) and lateral (**B**) views in a neonate with pulmonary atresia with intact ventricular septum. RV outflow tract ends blindly and the atretic pulmonary valve (APV) (thick arrows) is shown. The RV is moderately hypoplastic, but tripartite. Tricuspid regurgitation is noted with opacification of the right atrium (RA). The catheter (C) is curled in the RA; the balloon tip is marked with thin arrow in both (**A**) and (**B**). UAC, umbilical artery catheter; UVC, umbilical venous catheter. Reproduced from Rao, P.S. Neonatal Catheter Interventions. In *Cardiac Catheterization and Imaging (From Pediatrics to Geriatrics)*; Vijayalakshmi, I.B., Ed.; Jaypee Publications: New Delhi, India, 2015; pp. 388–432 [116].

**Figure 15 children-06-00054-f015:**
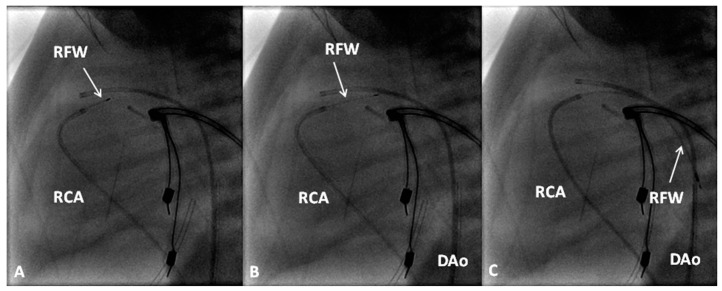
Selected cine frames in straight lateral views showing the radiofrequency wire (RFW) (thin arrows) progressively advanced from the right coronary artery (RCA) catheter into the main pulmonary artery (**A**) and (**B**) and from there it was further advanced into the descending aorta (DAo) in (**C**). Reproduced from Rao, P.S. Neonatal Catheter Interventions. In *Cardiac Catheterization and Imaging (From Pediatrics to Geriatrics)*; Vijayalakshmi, I.B., Ed.; Jaypee Publications: New Delhi, India, 2015; pp. 388–432 [116].

**Figure 16 children-06-00054-f016:**
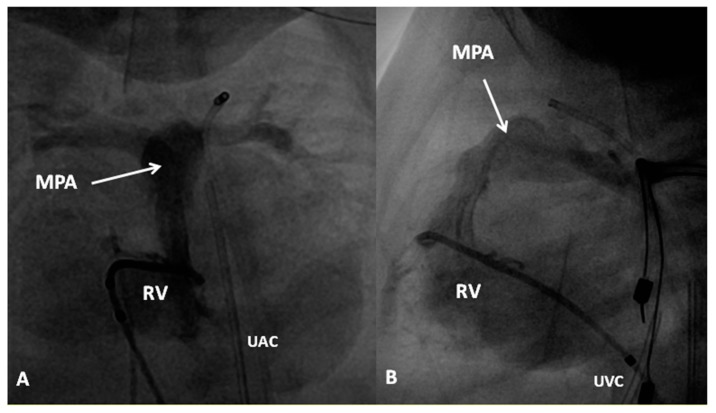
Selected cine frame from right ventricular (RV) cineangiogram in a sitting-up (15° LAO and 35° cranial) (**A**) and straight lateral (**B**) views immediately after radiofrequency perforation and balloon pulmonary valvuloplasty demonstrates prompt opacification of the main pulmonary artery (MPA). UAC, umbilical artery catheter; UVC, umbilical venous catheter. Reproduced from Rao, P.S. Neonatal Catheter Interventions. In *Cardiac Catheterization and Imaging (From Pediatrics to Geriatrics)*; Vijayalakshmi, I.B., Ed.; Jaypee Publications: New Delhi, India, 2015; pp. 388–432 [116].

**Figure 17 children-06-00054-f017:**
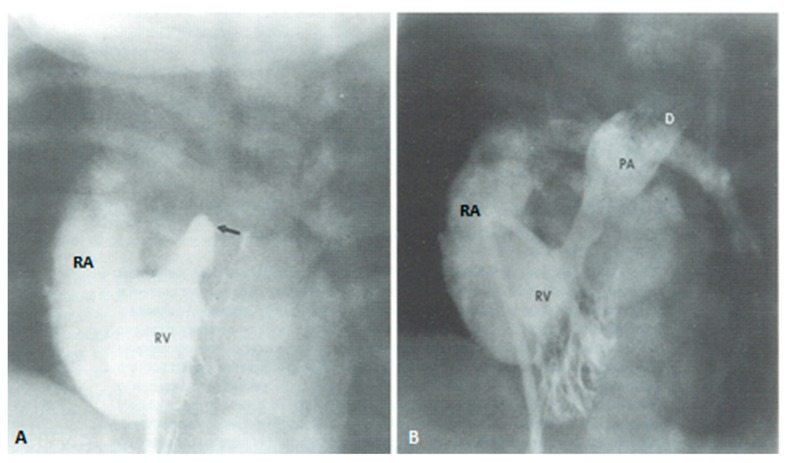
Selected frames from right ventricular cine-angiograms in a sitting-up view performed prior to (**A**) and 15 min following (**B**) perforation of pulmonary valve membrane and balloon pulmonary valvuloplasty. Note the small, heavily trabeculated right ventricle (RV) in (A) without anterograde opacification of the pulmonary artery. The arrow points to the atretic pulmonary valve membrane. Following the procedure, the pulmonary artery (PA) and its branches were well-opacified. Additionally, note opacification of pulmonary end of patent ductus arteriosus (D). There was significant tricuspid insufficiency opacifying the right atrium (RA) both before and after the procedure. Reproduced from Siblini, G.; Rao, P.S.; et al. *Cathet. Cardiovasc. Diagn.*
**1997**, *42*, 395–402 [24].

**Table 1 children-06-00054-t001:** Common cyanotic congenital heart defects.

5 Ts	Other Defects
Tetralogy of Fallot	Hypoplastic left heart syndrome
Transposition of the great arteries	Pulmonary atresia with intact ventricular septum
Tricuspid atresia	Double-outlet right ventricle
Total anomalous pulmonary venous connection	Double-inlet left ventricle and univentricular hearts
Truncus arteriosus	

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
