# Peer review of "Management of Congenital Heart Disease: State of the Art—Part II—Cyanotic Heart Defects"

_children, 2019, doi:10.3390/children6040054_

Round 1
Reviewer 1 Report
This review article is well-written, and typical clinical course and treatment regarding neonates with cyanotic heart defects are systematically described.
It may be informative if the author adds the relationship between chromosomal abnormalities and cyanotic heart defects to this manuscript.
Author Response
This review article is well-written, and typical clinical course and treatment regarding neonates with cyanotic heart defects are systematically described.
Thanks.
It may be informative if the author adds the relationship between chromosomal abnormalities and cyanotic heart defects to this manuscript.
Since the objectives of this review is to address the management of cyanotic heart defects, I respectfully decline to accept this recommendation.
Reviewer 2 Report
Thank you for the opportunity to review this article which appears to be the second part of a Review on the state of the art of management of congenital heart disease (CHD). The focus is here on cyanotic CHD, following part 1 on acyanotic CHD.
Overall the article is clear and well written, it provides a general introduction to the main scenarios in cyanotic CHD, with insights into the pathophysiology and main approaches for repair/palliation.
The article is clearly based on the author's own personal experience in the field. This is absolutely of interest but, as a general comment, I find a contradiction in the aim of describing the state of the art/full literature review. 70 references out of 129, at least, are from the author's own work, which clearly is relevant, but particularly in a complex field such as CHD where there are also differences between centres, historical differences in performing procedures, small case series etc this represents a major bias. I am not in any way doubting the quality of the work, but I think the article should be presented as "A single centre experience" or something along these lines, while covering the main cyanotic CHDs at it does. Otherwise, as a literature review, the article is not adequately citing important literature.
Other points:
- p.5: the discussion of Tet spells could perhaps be better integrated in the earlier description of ToF physiology; following from this, I wonder if also a paragraph on the main physiological features of cyanotic CHD broadly could be incorporated in the beginning of the manuscript, with main concepts applying pretty much to all scenarios being discussed, and then including specific points under each pathology
- p.12 and also 21: the Fontan operation is palliative and not curative, so perhaps the sub-heading 'Corrective surgery' is misleading, I would emphasise that for single ventricle physiology, within cyanotic CHD, there is a palliative but not curative approach, this concept should be made very clear especially for the less experienced readers (those seeking an overview of CHD by reading this review, this is a key concept)
- p.15: going back to presenting the full picture vs. personal experience, several centres also use MRI to evaluate their Fontan patients, rather than catheterisation, so if the article aims to comprehensively describe the state of the art the role of MRI is CHD is grossly underestimated in my opinion
- p.16: I would clarify that the fenestration is not necessarily always close at 6-12 months
- p.19 line 569: The PVR is higher, I believe this sentence needs revising, as indeed it is then discussed as 'PVR begins to fall'
- p. 23: Stage III is Fontan completion, I think Fontan conversion could be confusing terminology (in relation to failing Fontan).
Author Response
Thank you for the opportunity to review this article which appears to be the second part of a Review on the state of the art of management of congenital heart disease (CHD). The focus is here on cyanotic CHD, following part 1 on acyanotic CHD.
Agree and thanks
Overall the article is clear and well written, it provides a general introduction to the main scenarios in cyanotic CHD, with insights into the pathophysiology and main approaches for repair/palliation.
Thanks
The article is clearly based on the author's own personal experience in the field. This is absolutely of interest but, as a general comment, I find a contradiction in the aim of describing the state of the art/full literature review. 70 references out of 129, at least, are from the author's own work, which clearly is relevant, but particularly in a complex field such as CHD where there are also differences between centres, historical differences in performing procedures, small case series etc this represents a major bias. I am not in any way doubting the quality of the work, but I think the article should be presented as "A single centre experience" or something along these lines, while covering the main cyanotic CHDs at it does. Otherwise, as a literature review, the article is not adequately citing important literature.
As per the reviewer's suggestion, additional papers from the literature, as appropriate are cited in the revised script.
Other points:
- p.5: the discussion of Tet spells could perhaps be better integrated in the earlier description of ToF physiology; following from this, I wonder if also a paragraph on the main physiological features of cyanotic CHD broadly could be incorporated in the beginning of the manuscript, with main concepts applying pretty much to all scenarios being discussed, and then including specific points under each pathology
Tet spells are description in the early section as suggested. The main pathophysiologic features of cyanotic CHD are now added, as recommended.
- p.12 and also 21: the Fontan operation is palliative and not curative, so perhaps the sub-heading 'Corrective surgery' is misleading, I would emphasise that for single ventricle physiology, within cyanotic CHD, there is a palliative but not curative approach, this concept should be made very clear especially for the less experienced readers (those seeking an overview of CHD by reading this review, this is a key concept)
Agree; I have added a sentence to emphasize Fontan as palliative rather than corrective surgery.
- p.15: going back to presenting the full picture vs. personal experience, several centres also use MRI to evaluate their Fontan patients, rather than catheterisation, so if the article aims to comprehensively describe the state of the art the role of MRI is CHD is grossly underestimated in my opinion
The use MRI to evaluate their Fontan patients is mentioned as suggested.
- p.16: I would clarify that the fenestration is not necessarily always close at 6-12 months
Yes, the script is modified accordingly.
- p.19 line 569: The PVR is higher, I believe this sentence needs revising, as indeed it is then discussed as 'PVR begins to fall'
Changed as suggested.
- p. 23: Stage III is Fontan completion, I think Fontan conversion could be confusing terminology (in relation to failing Fontan).
Agree; I changed Fontan conversion to Fontan completion.
Reviewer 3 Report
This is a very comprehensive, well-structured, and well-referenced review on the management of CHD. Below are my suggestions:
include a definition of Cyanotic Heart Defects in the introduction.
include an index in the very beginning to help readers navigate through different sections and sub-sections.
Author Response
This is a very comprehensive, well-structured, and well-referenced review on the management of CHD. Below are my suggestions:
Thanks
include a definition of Cyanotic Heart Defects in the introduction.
Done; definition of cyanotic Heart Defects is included in the introduction section.
include an index in the very beginning to help readers navigate through different sections and sub-sections.
Done; a table indexing the defects is included in the introduction section.